# Bridging the Finger-Action Gap between Hand Patients and Healthy People in Daily Life with a Biomimetic System

**DOI:** 10.3390/biomimetics8010076

**Published:** 2023-02-11

**Authors:** Jong-Chen Chen

**Affiliations:** Information Management Department, National Yunlin University of Science and Technology, Yunlin 64002, Taiwan; jcchen@yuntech.edu.tw

**Keywords:** robotic control, computational intelligence, evolutionary learning, self-organizing learning

## Abstract

The hand is involved very deeply in our lives in daily activities. When a person loses some hand function, their life can be greatly affected. The use of robotic rehabilitation to assist patients in performing daily actions might help alleviate this problem. However, how to meet individual needs is a major problem in the application of robotic rehabilitation. A biomimetic system (artificial neuromolecular system, ANM) implemented on a digital machine is proposed to deal with the above problems. Two important biological features (structure–function relationship and evolutionary friendliness) are incorporated into this system. With these two important features, the ANM system can be shaped to meet the specific needs of each individual. In this study, the ANM system is used to help patients with different needs perform 8 actions similar to those that people use in everyday life. The data source of this study is our previous research results (data of 30 healthy people and 4 hand patients performing 8 activities of daily life). The results show that while each patient’s hand problem is different, the ANM can successfully translate each patient’s hand posture into normal human motion. In addition, the system can respond to this difference smoothly rather than dramatically when the patient’s hand motions vary both temporally (finger motion sequence) and spatially (finger curvature).

## 1. Introduction

The hand, a delicate design, is one of the most important structures of the human body. Whether its function can work normally will greatly affect our quality of life. However, in the real world, some people may lose part of their finger functions due to external factors (such as work accidents, car accidents, surgery) or internal problems (such as aging, genetic diseases, infections, strokes), which might affect the quality of life of patients. How to help these people regain their normal activities has become a very important issue. For patients, proper and timely rehabilitation plays a key factor. However, the loss of function is different for each hand patient, resulting in different rehabilitation needs for each individual. Even the same patient may need different rehabilitation treatments at different times. Facing this problem, how to meet the rehabilitation needs of each individual is a large challenge. Without a doubt, human-assisted patient rehabilitation is one of the best solutions. However, this method still has its inherent disadvantages, such as considerable labor costs and the inevitable possibility of human negligence. Robots, with motion controllability and measurement reliability, can more or less help to solve some of the above problems [1].

An analytical review of the literature on the advantages and disadvantages of using robots for rehabilitation can be found in [2,3,4]. There is a considerable amount of research on the use of robotics in rehabilitation. Some studies focus on utilizing robots for upper-body daily activities, while others focus on lower-body walking gaits. The following literature review only highlights relevant research on data gloves (or robotic fingers) and hand exoskeleton devices (or prosthetic hands). Many researchers have developed their data gloves to collect and analyze data on people’s fingers in various everyday activities [5,6,7,8,9,10,11]. These data are then used to design relevant assistive robots. Unfortunately, there is currently no unanimous set of rules and procedures for developing these data gloves. The main difference between them is not only the difference in the sensor materials used but also the difference in the functions emphasized. In addition, the number of sensors used in each glove is also different. Some scholars designed a data glove with several times more sensors than any of the above studies and applied it to explore the characteristics of human grasping [12]. On the contrary, some other scholars use relatively few sensors to achieve the same data collection results [13]. In general, too few sensors have the problem of collecting insufficient data. On the contrary, if there are too many, it may have problems such as mutual interference of sensors and insufficient space for sensors. In general, the location of a sensor may greatly affect the accuracy of its collected data. In response to this problem, some scholars try to optimize the arrangement of sensors in different ways [14]. Not only does each glove use a different number of sensors, but the placement of the sensors can also vary widely. Every data glove has its specific design and features and, of course, has its pros and cons. The above studies emphasize the use of data gloves for data collection and analysis. In addition, some other studies have emphasized its application in clinical research [15,16,17].

In the field of finger-controlled robotics research, there are currently many types of robotic arms with grasping (gripping) capabilities [18]. If divided from the source of power, it includes vacuum, pneumatic, electric, and hydraulic. In addition to the different power sources mentioned above, the arms may also have different numbers of jaws. At present, most robots are designed with hard materials, but there are still some robotic arms that choose to use relatively soft materials [19]. Moreover, generally speaking, the control of the movement of the robotic fingers is accomplished through close contact, but there are also a group of scholars who control the movement of the fingers by remote control [20]. Each design has special features suitable for the various fields of application and different needs.

In the field of robotic rehabilitation, another line of scholars starts from the direction of the exoskeleton of the hand (or prosthetic hand). Research in this line hopes to integrate advanced technologies in different fields (such as sensors, actuators, mechanical structures, motion theory, artificial intelligence, algorithms, and control) to assist impaired patients to restore their motor ability [21,22,23]. However, these advanced devices are designed with pre-set ideas and thinking, or it has been pre-determined to solve some specific problems [24]. However, as previously mentioned, the rehabilitation needs of each hand patient at each stage are different. Even more challenging is that in some cases, regardless of how hard patients try, they are still unable to perform movements that healthy people can easily perform in daily life. In such a case, the role that rehabilitation can play is limited. How to bridge the finger-action gap between hand patients and healthy people in daily life is a major issue. It is, thus, the software system that might play a very important role. Unfortunately, the development of today’s software systems is still based on how to emphasize the direction of programmatic design. In this direction, people will be committed to using symbols to represent people and things in computers, and then use some symbol manipulation methods (i.e., algorithms) to solve specific problems. The problem challenging the software world is very similar to that of hardware (i.e., it is difficult to make specific adjustments to a problem). Faced with this problem, artificial intelligence (or the so-called soft computing) acts as a bridge to deal with some of these problems.

This study hopes to establish a system with an autonomous learning capability that can provide some kind of assistance to users with special needs in a self-adjusting way. The ANM system is a biologically motivated system that captures the close structure/function relationship of biological systems [25,26]. This research team has previously successfully demonstrated that it has sufficient autonomous learning ability to learn to control the walking of a quadruped robot [27], the serpentine motion of a snake-like robot [28], and the human-like rehabilitation movement of a robot arm [29]. These actions are mainly achieved by adjusting and controlling the rotation angle and starting time of different motors by the ANM system. Unlike the above-mentioned research, this study attempts to superimpose the output signals of the ANM system to generate a wave pattern that controls finger bending. Our goal is to bridge the gap in daily life finger movements between hand patients and healthy individuals through the self-learning capabilities of the ANN system that we demonstrated previously (i.e., building a customizable prosthetic finger control system). However, it should be added here that the current research stage of this study is that the control of the prosthetic finger system is verified under a simulated platform, rather than under a physical operating environment.

## 2. Materials and Methods

### 2.1. The Architecture of the ANM System

As mentioned above, the purpose of the whole study is to build a bridge between the actions of patients and healthy people. To achieve this goal, the ANM system must be able to make appropriate adjustments according to individual needs (that is, transduce the patient’s actions into healthy people’s actions). This adjustment is accomplished through two types of neurons with different functions. One type is a neuron with a gradual transformation function (called information-processing neurons, IP neurons), and the other type is a neuron with a function of controlling (selecting) other neurons (called control neurons, CN neurons). The former is responsible for transducing a series of spatiotemporal signals into another series of spatiotemporal signals, while the latter selects appropriate information-processing neurons that engage in signal transduction. Only information-processing neurons selected by control neurons are allowed to engage in input/output transduction. The ANM system combines these two types of neurons into a single, closely integrated architecture (Figure 1).

### 2.2. Information Processing Mechanism of an IP Neuron

The information processing mechanism of IP neurons is motivated by two physiological hypotheses (evidence) [30,31,32,33]. One is that a neuron’s internal dynamics control its firing behavior. The other hypothesis is that the cytoskeleton may be where information processing occurs. Physiological evidence shows that the cytoskeleton is composed of many different types of components (i.e., microtubules, microfilaments, neurofilaments), which are connected by microtubule- or neurofilament-associated proteins (in this study, we collectively refer to these proteins as MAPs). In the ANM system, this study hypothesizes that the cytoskeleton plays a role in integrating different spatiotemporal signals and transforming them into a cascade of spatiotemporal signals. The cytoskeleton is abstracted as a two-dimensional grid structure (Figure 2). The intra-neuronal dynamics of cytoskeletal neurons are simulated with two-dimensional cellular automata. The information processing about the cytoskeleton in this study is based on three assumptions. The first is that the entire cytoskeleton has three different types of basic constituent molecules that play the role of signal transmission. Each type of molecule has a different transmission speed and different degrees of influence on each other. The second assumption is that the system has two types of enzymes (readin and readout) that serve as the input–output interface of the cytoskeleton. The third assumption is that different signals can influence each other through MAPs.

The whole operation process is as follows. When an external signal is sent to a readin enzyme on the cytoskeleton of a neuron, it will activate the cytoskeletal component at the same location (note: if there is no molecule in that place, the signal activated by the readin enzyme is lost). The activated cytoskeleton component will further activate its adjacent components of the same type (note: adjacent cytoskeleton components of different types will not be activated). By analogy, it will, in turn, activate the same type of cytoskeletal components (i.e., generate a signal flow). For example, when an external signal is sent to the readin enzyme at location (2, 4), a signal flow of the c3 components starting from location (2, 4) to location (8, 4) will be activated. However, the above-mentioned signal flow will not have any influence on any other adjacent cytoskeletal components of different types. However, if there is a MAP that links two different types of cytoskeletal components, it will have some degree of influence (note: the degree of influence is asymmetric). Therefore, if the c1 component at (4, 7) is activated, it can affect the c3 component at (4, 8) through MAP. Similarly, if the c3 component at (4, 8) is activated, it can affect the c1 component at (4, 1) through the MAP connecting them. However, the former has a greater influence on the latter (that is, it is easy to stimulate it to generate signal flow). In contrast, the latter only increases its activation of the former to a certain extent. The above assumption is that the interaction between two adjacent components of different types is asymmetric. In addition, we assume that different types of components transmit signals at different speeds. In summary, one type of element transmits at the slowest speed but has the highest activation value for other types of cytoskeletal components. The other transmits the signal at the fastest speed but has the lowest activation value for other types of cytoskeletal components. The transmission speed and activation value of the third type of component are between the former two. When a cytoskeletal signal passes through a readout enzyme, it changes the activation state of that enzyme. If another type of signal affects the readout enzyme through the MAP for a short period, the neuron will fire. Of course, if an activated readout enzyme is not stimulated for a period, its activation state will gradually decay over time. Before any learning, the cytoskeleton configuration of each IP neuron is randomly determined. Because of this, each neuron differs in how its initial signals flow, how they interact with each other, and how they receive input and output signals. Through evolutionary learning, the above settings will be changed accordingly (the details are explained in Section 2.5).

### 2.3. Information Processing Networks

The previous section introduced the operation mechanism of an IP neuron. In the following, we introduce the entire network and its relationship with each other. In the current implementation, this study assumes the system is comprised of 400 IP neurons. These neurons are equally divided into eight subnets (each subnet consists of 50 neurons), forming a competing network. Each of the eight subnets has a neuron with similar intra-neuron and inter-neuron relationships (we call it a bundle), as described above. These 400 IP neurons can be regarded as 50 bundles if we divide them from the perspective of the internal structure and external connections of similar neurons across subnetworks. From the perspective of evolutionary learning (selection, replication, mutation), these are a group of neurons with similar intra-neuron structures and inter-neuron connection patterns. All evolutionary learning occurs only in different subnets of neurons belonging to the same bundle. This feature is important for facilitating evolutionary learning. As mentioned earlier, one of our goals in the system is to capture biological-like structure/function relationships. That is, when the structure of a system changes slightly, its function (or behavior) changes gradually. With this property, given the same input, neurons with similar intra-neuron structures should behave similarly.

### 2.4. Control Networks

Although current technology is quite developed and advanced, however, how we form and store memories is still a mystery that has not been fully solved. It is generally believed that the hippopotamus is the main location of long-term memory. The control structure of this study is motivated by the parallel, sequential, and hierarchical structure of the hippopotamus memory structure. Some related literature can also be found in [34]. In the ANM system, the CN neuron plays the role of controlling IP neurons to engage in input–output transduction. The control mechanism is that the synaptic connections between a reference neuron and cytoskeletal neurons are facilitated if they fire at the same time. The later firing of the reference neuron will cause all the cytoskeletal neurons controlled by it to fire. In Section 2.3, we already introduce the entire IP network, which can also be divided into 8 competitive subnets (or into 50 bundles of neurons). As mentioned above, this study assumes that not all IP neurons will participate in signal transduction (note: only IP controlled by CN neurons are allowed to participate). The entire IP network is controlled (selected) by two layers (high and low) of CN neurons. Each low-level CN neuron is responsible for controlling a bundle of neurons (this connection relationship is fixed so that it will not change as the number of learning times increases). Each high-level CN neuron is responsible for controlling (selecting) several low-level CN neurons (the selections are not fixed and will change as the number of learning times increases).

The two-level CN neurons, thus, form a hierarchical control architecture whereby activation of a higher-level CN neuron will fire all lower-level CN neurons controlled by it. In turn, the firing of one lower-level CN neuron will fire all IP neurons in the same bundle (i.e., neurons in different subnets have similar intra-neuronal structures and inter-neuronal connections). The connections between high-level reference neurons and low-level layers of reference neurons change during learning. Figure 3 provides a simplified picture (only two competing subnets are shown, each with 50 IP neurons).

### 2.5. Evolutionary Learning at the Level of IP Neurons

In the beginning, the internal structure of each IP neuron and the connection between neurons are randomly determined (the C1, C2, and C3 signal transmission components on the cytoskeleton, MAP, readin, readout, and the connection with the input relation). Each cycle of evolutionary learning repeats the following three steps until learning is stopped:Evaluate the suitability of each subnet;Select the subnet with better performance;Copy and mutate from a better-performing subnet to a poorer subnet. The copy and mutation step occurs between the same bundle of IP neurons (the copying and mutation of C1, C2, and C3 signal transmission components on the cytoskeleton, MAP, readin, readout, and pattern of connections with input).

### 2.6. Evolutionary Learning at the Level of CN Neurons

As described in Section 2.4., the ANM system controls IP neurons through two layers of CN neurons (that is, only neurons selected by CN neurons are allowed to process input/output information). Evolutionary learning occurs between high-level and low-level to generate different combinations of IP neurons (note: the connection relationship between low-level CN neurons and IP neurons will not change during learning). In the beginning, the low-level CN neurons selected by each high-level CN neuron are randomly determined. Each cycle of evolutionary learning repeats the following three steps until learning is stopped:Evaluate the fitness of IP neurons selected by high-level CN neurons (via low-level CN neurons);Select high-level CN neurons with better performance;Copy and make mutations change from high-level CN neurons with better performance to relatively poor high-level CN neurons. The copy and mutation step occurs in the combination of low-level CN neurons selected by high-level CN neurons.

## 3. Application Domain

In this section, we first explain the experimental daily actions that we selected. We then show how to link the inputs and outputs of the ANM system to the problem domain (hand movement data). Finally, we illustrate how to perform a fitness assessment.

### 3.1. Experimental Daily Actions

In this study, 8 of the 32 people’s daily life actions listed in [35] were selected as the experimental actions of this study, as shown in Figure 4. The first action is to make a gesture of holding a wine bottle but not touch any object (to be referred to as virtual bottle holding), while the other 7 actions have actual contact with objects. These seven actions are holding a wine bottle, holding a water bottle, holding a mug, squeezing toothpaste, holding a ping pong ball, holding a marble, and manipulating the mouse.

As mentioned earlier, the data used in this study were derived from the previous research of this team. In that study, we made an induction glove and then asked 30 people with healthy hand actions as well as 4 hand patients to perform the above 8 actions (note: these subjects were all right-handed). We add here that all data collection in that study was performed with care. Before the experiment and after a period of the experiment, the research team conducted a timely reliability and validity analysis on each sensor to ensure the accuracy of the data. The method we adopted was to use a protractor from 0 to 90, and then use 15 degrees as a measurement angle interval to make 6 angle lines. For each angle, we bent the camber sensor to align with the predetermined angle and calculated the difference. For each angle value, we repeated the test 5 times to ensure that the angle value collected by the curvature sensor was correct. However, it is emphasized here that the focus of this article is not to analyze the data of these patients, but to achieve the purpose of making actions similar to healthy people through autonomous learning. Therefore, this study here only hopes to present the data of the patients without further analysis of these data. If it can be proved that there are some differences in the data of these patients, the ANM system can be used to narrow this gap. This is one of the main purposes of using this data glove in this study.

Figure 5 shows the time-series data of 8 movements of five fingers of healthy hands. Figure 6 shows the value of the maximum curvature of each action during the action. From Figure 6, we can see that among the five fingers, the middle finger has the largest curvature value, the two adjacent fingers (index finger and ring finger) have the next largest, and the thumb and little finger have the smallest.

Figure 7 shows the time series data of 4 hand patients. Clinically, these four patients all had a history of “squeeze injury” in their hands. After a long period of doctor treatment and patients’ rehabilitation, some of their hands still had the sequelae of finger stiffness, meaning that these fingers cannot be fully bent or extended. These four patients had two issues. The first issue was that everyone’s five fingers had different degrees of stiffness. The second issue was that different actions had different requirements for the degree of finger bending. For example, it requires a relatively large curvature of the finger movement when people need to perform a rough grasping action (for example, holding a bottle). When these two issues are put together, we can see that each patient showed different finger movements for each action and the degree of finger stiffness was also quite varied (Figure 7). Taking virtual bottle holding as an example, the curvature of the fingers of the four patients varied considerably. For some patients (p1 and p2), the index finger and ring finger were even more curved than the middle finger (note: generally speaking, the middle finger of a healthy person is the most curved of the five fingers).

### 3.2. Input/Output Interface

The ANM system consists of 8 competing subnets (each has comparable intra-neuronal structures and inter-neuronal connections). Figure 8 shows the input/output interface of one of the eight subnets (the remaining 7 subnets have the same input/output interface). The main purpose of the whole system is to transduce the patient’s five-finger action data into five-finger action data, similar to that of healthy people. In terms of input, the time-series finger data of a certain movement of a patient will be sequentially fed to the ANM system at a fixed time interval (there are 50 records for each movement, and each record has the curvature data of 5 fingers). In terms of output, all IP neurons are divided into five categories, each corresponding to the control of a certain finger. A total of 5 temporal output data are required for five fingers.

### 3.3. Fitness Function

As mentioned above, each subnetwork of the ANM system has 50 IP neurons. These neurons are equally divided into five groups according to the classification of the five fingers. The firing behavior of each group of IP neurons represents the manipulation control of a certain finger. In the present implementation, we use the time difference between two adjacent firing neurons of the same group to represent the degree of finger curvature control. This study further hypothesizes a sigmoid-like waveform relationship between the time difference and the degree of finger curvature control. In other words, the value of the degree of finger curvature control increases exponentially with the increase in the time difference. After many attempts, we decide to use the formula shown in Equation (1). For example, suppose the time interval between two neuron firings is 0.051205 milliseconds. If we enter this value into Equation (1), we obtain a value of 4.6 (as shown in Equation (2)). If a new waveform is generated shortly after one waveform, the two waveforms will be processed in a tandem superimposed manner. A double-peak waveform is formed at the two peaks, and the overlapping area in the middle will be superimposed. The waveforms generated by the firing behavior of the neurons in the same group are superimposed to form a relatively large waveform (i.e., the action controlled by the curvature of the finger).
(1)$$\mathrm{the}\;\mathrm{degree}\;\mathrm{of}\;\mathrm{finger}\;\mathrm{curvature}\;\mathrm{control}=(\frac{1}{1+e\left(-2\times \Delta t\right)}\;-0.5)\;\times 2\times 90$$
(2)$$\mathrm{the}\;\mathrm{degree}\;\mathrm{of}\;\mathrm{finger}\;\mathrm{curvature}\;\mathrm{control}=(\frac{1}{1+e\left(-2\times 0.051205\right)}\;-0.5)\;\times 2\times 90=4.6$$

The ANM system superimposes (transduces) all the waveforms generated from the 50 input records into time-series data of hand activities. The final waveform (the time-series data) will be contrasted with the time-series data of healthy subjects. The fitness is measured as the difference (called loss) between these two time-series data, as shown in Equation (3). Loss represents the sum of all the curvature differences of all fingers of the patient and the healthy person during the action. The smaller the loss value, the better the fitness of the system. When the loss value is divided by 50-time points for 5 fingers, it represents the average value of the difference in finger curvature (to be referred to as $\bar{\mathrm{loss}}$), as shown in Equation (4).
(3)$$\mathrm{loss}=\underset{i}{\sum }\left|\overset{50}{\underset{j=1}{\sum }}\left(H_{\mathrm{ij}}-P_{\mathrm{ij}}\right)\right|$$
where $H_{\mathrm{ij}}$ and $P_{\mathrm{ij}}$ represent the curvature data of healthy and patient, respectively; I = thumb, index, middle, ring, or little finger.
(4)$$\bar{\mathrm{loss}}=\frac{\mathrm{loss}}{\left(50\times 5\right)}$$

## 4. Experiments

As mentioned earlier, the purpose of this study is to establish an autonomous learning platform to assist patients to make actions similar to healthy people. However, we note that each patient has different hand problems. Therefore, the learning platform must prove that the time series data of a patient can be successfully translated into those of healthy people. In other words, the platform must show that it can figure out how to translate between two different time series data through autonomous learning. At the present stage, it can be called a success if the system is capable of completing the above-mentioned transduction. Furthermore, humans are not machines. Even if it is the same action, people do it more or less differently every time. One is the difference in the degree of bending of each finger during actuation, and the other is the difference in the timing of finger movement. Therefore, we must further explore whether the output of the system shows a smooth or drastic change when this difference occurs. The following experiments are divided into two parts: adaptive learning and disturbance tolerance. Adaptive learning is to let the ANM system learn to transduce the actions of each patient into the actions of healthy people. Fault tolerance is to test the output of a long-term learning system in the face of different spatiotemporal disturbances to previously learned data.

### 4.1. Adaptive Learning

We have shown the data of 8 actions of 4 patients in Figure 7. For each patient’s action, the system must learn to convert it into the data of people with healthy hands (Figure 5). A total of 32 experiments were performed for 4 patients and 8 actions. Each experiment was performed independently. Figure 9 shows the learning results for each experiment. It shows that the system can reduce the loss value to a fairly low level. The learning process showed that the system can greatly reduce its loss value in the early learning stage and become comparatively slow as the number of learning times increases. For example, as shown in Figure 9, in the early stages of learning, the system can reduce the loss value by about 80% in less than 10 rounds of learning (note: each round is 32 learning cycles in the system). In contrast, in the later stages of learning, the system may need many more rounds of learning to reduce the loss value a little. However, the most important thing is that it does not show the stagnation of learning. The findings of this study show that when the system is given a sufficiently long number of learning times, it can still exhibit progressive learning. In other words, if we continue to extend the learning time, the ANM system has the opportunity to move closer to completing the assigned task.

The results showed that regardless of which of the eight movements, the ANM system is able to successfully translate each patient’s hand movements into movements close to those of healthy people after a period of training. Taking patient 1’s virtual bottle holding action as an example, its initial $\bar{\mathrm{loss}}$ value is 27.0 degrees. This means that the curvature of each finger differs by 27 degrees at each time point between patient 1 and the healthy subjects. In other words, the movement of patient 1 is quite different from that of the healthy subjects (the average difference is 27 degrees). With the help of the ANM system, the $\bar{\mathrm{loss}}$ value can be reduced to 3.9 degrees (85% improvement). Through the help of the ANM system, we can reduce this difference to 3.9. In other words, if patient 1′s finger actions are assisted by the control of the ANM system, it can make actions close to those of a healthy person. Likewise, with the assisted control of the ANM system, the difference between patient 2 and the healthy subjects could be reduced from 22.8 degrees to 5.3 degrees, patient 3 from 22.0 degrees to 3.5 degrees, and patient 4 from 27.0 degrees to 4.0 degrees. As shown in Table 1, the result shows that each patient can perform actions close to those of healthy people with the assistance of the ANM system control (the average difference in curvature of each patient from healthy people is within 5 degrees).

Table 2 shows the difference between patients (including between each patient and healthy people) before learning using the ANN system. From Table 2, we can see that the gap between them (in terms of $\bar{\mathrm{loss}}$ value) is quite large. However, after learning, the difference between patients and healthy people significantly improves (Table 3). While the gap between patients does not improve as much as the gap between patients and healthy people, it does show some improvement. This result is that ANM systems are designed to improve the gap between each patient and healthy people (not between patients). However, each patient still has its sequelae that are different from others. In this way, they will all improve toward the same learning goal, but there is no doubt that each still has its problems.

Figure 10 shows the actuation of five fingers per patient (time series output) with the assistance of the ANM system. That is, after the conversion of the ANM system, the general results of each finger operation of each patient are very similar. This result proves that the ANM system can successfully transduce each of the original movements of the four patients into the movements of healthy people.

To understand the learning performance of the ANN system, the work of this study is further compared with one of state-of-art systems learning systems—MLP (multilayer perceptron). Table 4 shows the comparison of the learning results of the ANN system and MLP. It shows that in a total of 32 cases (4 patients with 8 actions), there are 21 cases that the ANN system performs with a better learning effect than MLP (about 67.7%). We noted there are still 11 out of these 32 cases (about 32.3%) where MLP has a better learning performance than the ANN system. As most of the neurons in the ANN system have internal dynamic processing functions, the amount of computational time required to simulate this type of data processing with a digital computer is relatively large (especially using a sequential processing machine to simulate a parallel processing system). This greatly limits the number of neurons that the ANN system can currently simulate, resulting in some cases where its performance is not as good as MLP. In contrast, MLP (or other state-of-art systems) can use relatively large networks for information processing because there is no such limitation. When more neurons with intrinsic motivation are allowed in the ANM system, its information processing repertoire will increase accordingly. We add that the ANM system has been successfully demonstrated in different, previous experimental domains where it had the capability of continuous learning [27,28]. As a consequence, when the processing speed for simulating these operations is significantly improved in the future, ANN systems can most likely perform better than MLPs by allowing more neurons with intrinsic dynamics.

### 4.2. Noise Tolerance

We all know that in real life, no one performs the same action the same every time. More precisely, the degree of curvature of the fingers and the sequence of finger actions will be more or less changed each time. In dealing with this problem, the approach adopted in this study is to use the ANM system after substantial learning, then make a series of changes to each action data of each patient, and compare the results under different degrees of change. It should be noted that the so-called serial change means that in the degree of change, we adopt a method of gradually increasing the range, but the real value is generated randomly. The ranges of change currently used are 1, 2, 3, 4, 5, 10, 15, 20, 25, 30, 35, 40, 45, and 50%. The so-called random here refers to using a random number to determine whether to change the size and order of the curvature of the fingers within the specified range of variation. In addition, the sequence of random numbers used is also randomly determined. In this case, although the magnitude of the change is the main factor of the change, the results are not proportional. However, we can be sure that when the number of tests increases, the results are positively correlated.

The results (Figure 11) show that the loss value of the ANM system varies correspondingly as the degree of patient action changes increases. However, it must be emphasized that the rate of increase does not present a steep change, but a near-smooth increase. Among the eight actions, when compared with the other five, the three actions (holding a mug, holding an empty wine bottle, and squeezing toothpaste) increase faster. This is because these three actions generally require a relatively large finger bend, especially when holding a mug (sometimes the finger bend is required to exceed 90 degrees). For patients, these actions are relatively difficult to achieve. Thus, they are more sensitive to noise. In addition, grasping marbles and table tennis are two that require relatively low curvature. Correspondingly, the sensitivity to noise is also low. For patients, these actions are also relatively difficult to achieve because of the large curvature requirements of the action. Therefore, their sensitivity to motion disturbances is relatively high. In contrast, catching marbles and table tennis are two actions that have relatively low curvature requirements, so they are relatively less sensitive to noise. Based on the above results, the ANM system used in this study can successfully transduce the patient’s actions into those of healthy people. As the degree of noise increases, the output of the system (loss value) also changes smoothly (that is, there is a gradually changing relationship between the two).

## 5. Discussion

In daily life, a pair of healthy hands allows us to easily and naturally complete the actions we want. However, we can only truly understand how inconvenient it is when people lose the function of their hands, even if only a small part. How to help these people to recover hand function as much as possible is an important issue. In recent years, advances in kinematics, motion theory, sensors, and artificial intelligence algorithms have brought hope for people to use hand-assistive devices to restore the motor ability of impaired patients. If the above technologies are well elaborated together, it may be possible to meet the needs of some patients. However, how to design a software and hardware system to meet the needs of different patients at different times is an open issue.

In this study, the four patients all have a history of clinically so-called “crushing injury” in the hand. Each of them had different hand injuries and sequelae and, therefore, considerable variation in the degree to which the fingers were bent. These patients all have their needs for using machines to assist individuals, and how to meet these needs is a problem that cannot be solved by sophisticated design in advance. We note that the data used in this study were not collected through very high-precision instruments. Without a doubt, the data themself cannot 100% reflect the real hand information of the patients. However, the purpose of this study is not to apply it to a patient immediately, but to demonstrate its ability to transduce one time-series data point into another time-series data point. Under such considerations, the authenticity of data may not be the focus of the discussion, but rather whether a successful transduction mechanism exists. This study calls it a “success” if appropriate transduction can be made under any two different sets of time series data. Furthermore, the ultimate goal of this research is not only to close the gap between the movement of patients and healthy people but also to hope that this gap can change smoothly as the movement of patients varies. Under this premise, when discussing the pros and cons of a system, the combination of different time series should be considered instead of judging only by the results of a single time series. In response to this problem, the current system performance is analyzed under the assumption that the data are disturbed in different degrees of time and space. Therefore, this study designs two different experiments, one of which is about adaptive learning and the other about fault tolerance. According to the author’s understanding, the current research in this direction is quite limited.

In this study, the ANM system with autonomous learning capability is used to successfully transduce each patient’s movements into those of healthy people. This means that the system has a customization function, which can be used to generate the necessary input/output conversion according to individual needs. It must be emphasized again here that the entire non-linear conversion mechanism is completed by the system in a self-changing manner. Furthermore, this study not only shows that the learning curve of the whole system presents a smooth improvement method, but also, more importantly, its learning presents a situation of continuous improvement (there is no phenomenon of complete stagnation of learning). This result suggests that when we allow the system to learn long enough, it can move toward complete problem-solving.

In addition to proving that the system can meet individual needs (customization), a very important point is that it must be capable of dealing with noise. In other words, when the patient’s motion changes slightly, the output of the system can show relative changes (rather than produce drastic changes). This study explored this issue by varying the magnitude of finger flexion and the timing of finger actuation in patients. The results show that the system can gradually change as the level of disturbance increases.

## 6. Conclusions

The rich dynamics of the ANM system used in this study enable us to use evolutionary learning mechanisms to deal with relatively complex problems. The richness of the system is based on the internal dynamics of neurons and is achieved in a functionally complementary manner through two types of neurons with different characteristics (information processing and control neurons). In terms of the internal dynamics of the system, the ANM system is completed through three elements: weak interaction, redundancy, and compartmentalization [25]. In addition, the multilayered structure of the system makes it possible to experiment with the interplay of evolutionary changes at different levels [26] (i.e., evolution at one level opens up possibilities for evolution at other levels and vice versa). Such interactions are a universal feature of living things and other complex systems.

This study is currently only a preliminary study of finger curvature. There is no doubt that it is not fully mature. In the future, this study should recruit more participants with different types of hand injuries to explore the ability of system customization. In addition to curvature research, acupressure can also be considered in the future to meet more clinical needs and develop a wider application space.

## Figures and Tables

**Figure 1 biomimetics-08-00076-f001:**
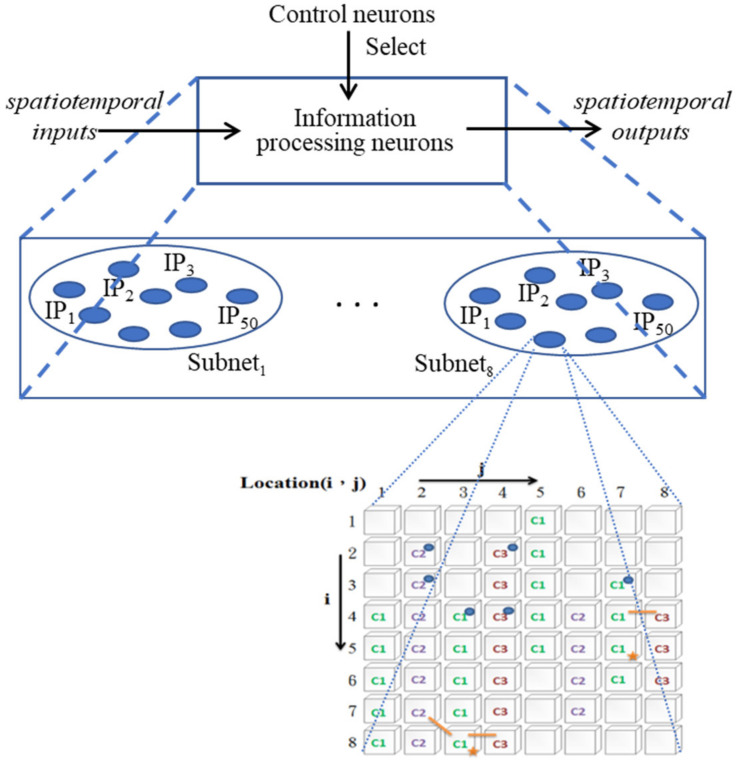
The architecture of the ANM system.

**Figure 2 biomimetics-08-00076-f002:**
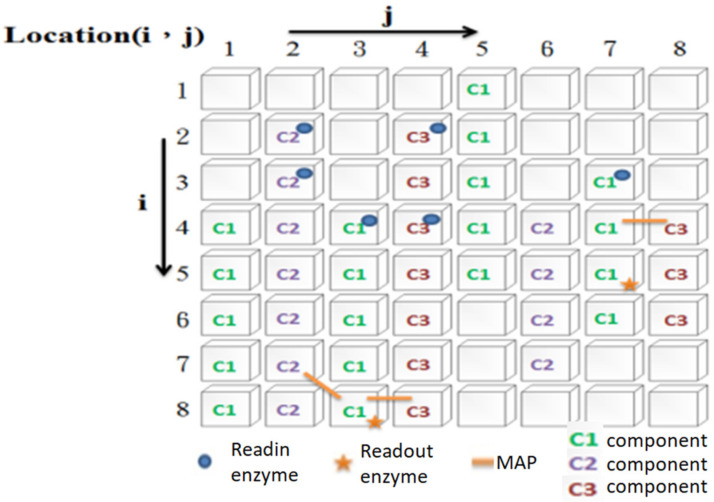
A simulated two-dimensional grid structure of an IP neuron.

**Figure 3 biomimetics-08-00076-f003:**
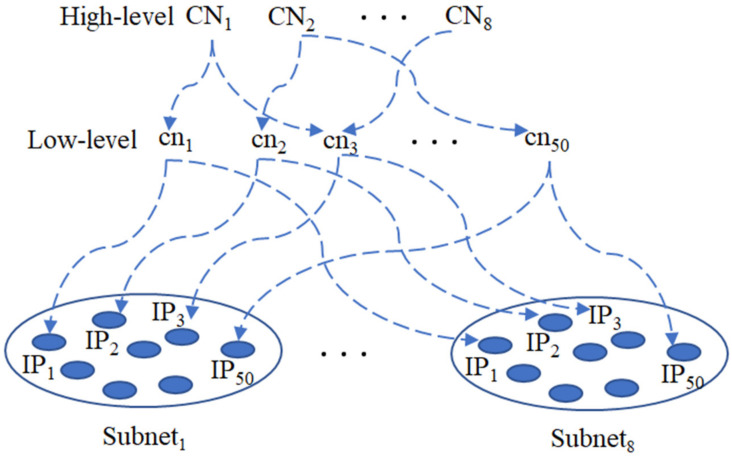
Two-level hierarchical control architecture.

**Figure 4 biomimetics-08-00076-f004:**
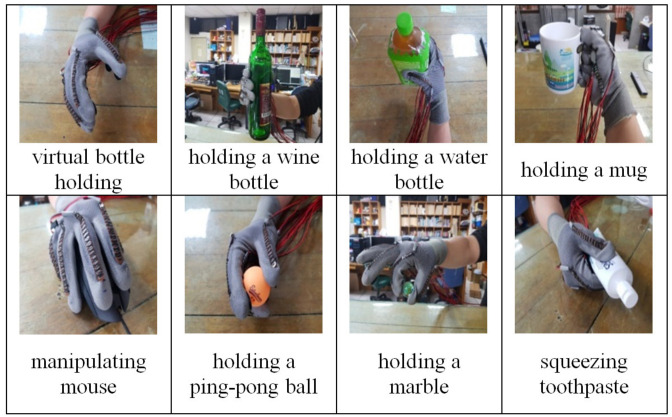
Eight daily actions.

**Figure 5 biomimetics-08-00076-f005:**
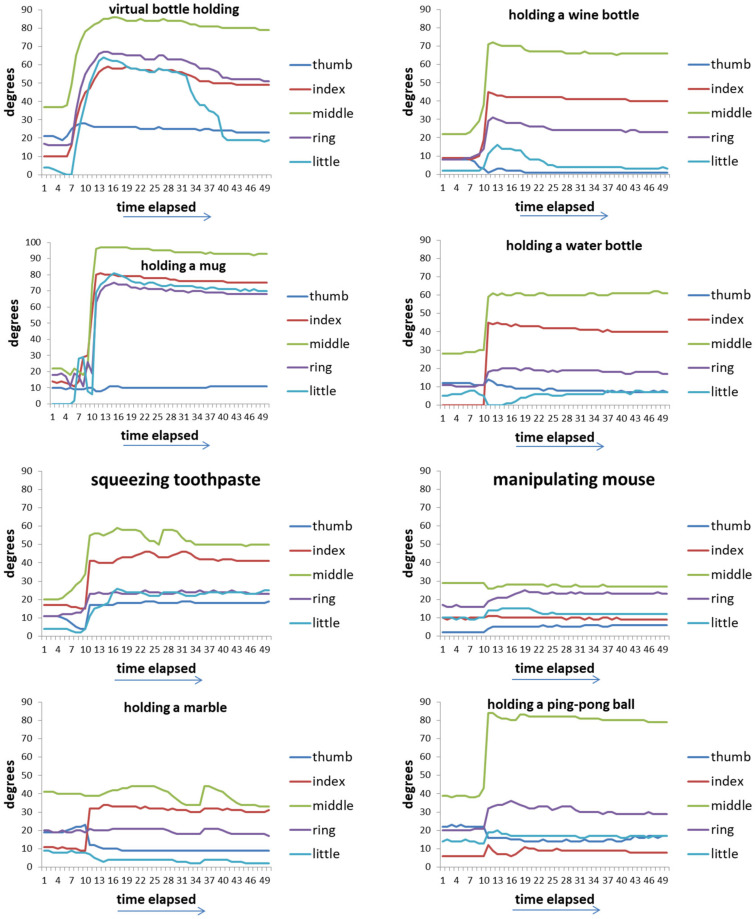
Time-series data of 8 movements of five fingers in each movement of healthy hands.

**Figure 6 biomimetics-08-00076-f006:**
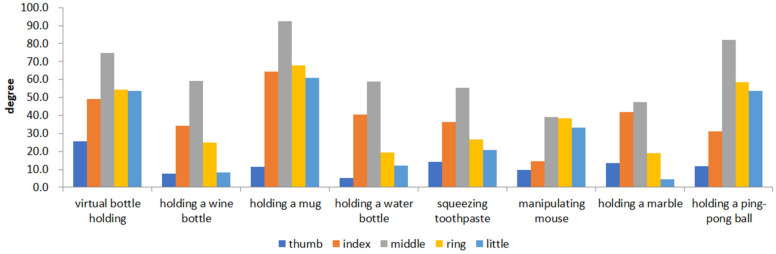
The maximum curvature values of five fingers of healthy hands.

**Figure 7 biomimetics-08-00076-f007:**
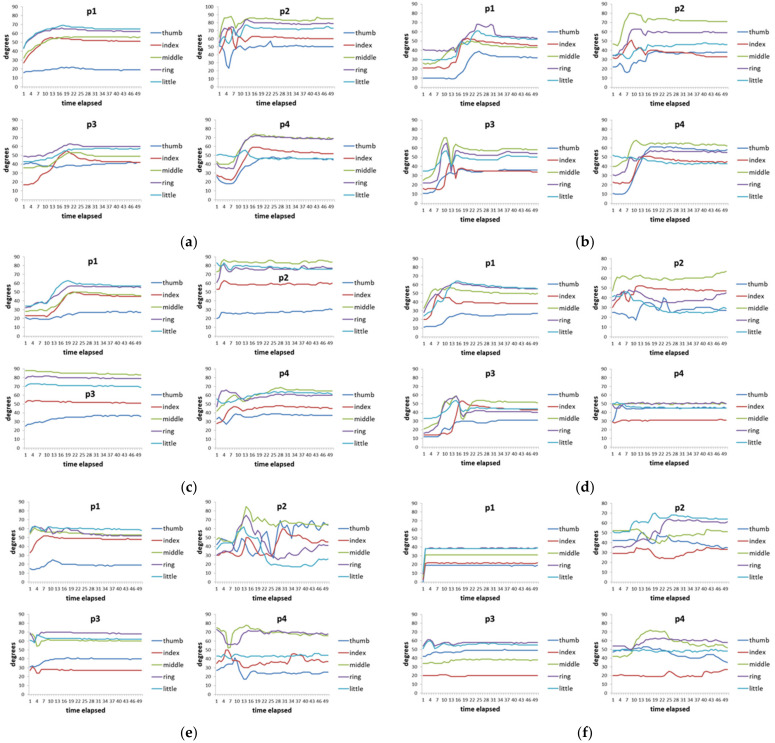
Time-series data of 8 actions of 4 hand patients. (**a**) Virtual bottle holding; (**b**) holding a wine bottle; (**c**) holding a mug; (**d**) holding a water bottle, (**e**) squeezing toothpaste; (**f**) manipulating mouse; (**g**) holding a marble; (**h**) holding a ping-pong ball.

**Figure 8 biomimetics-08-00076-f008:**
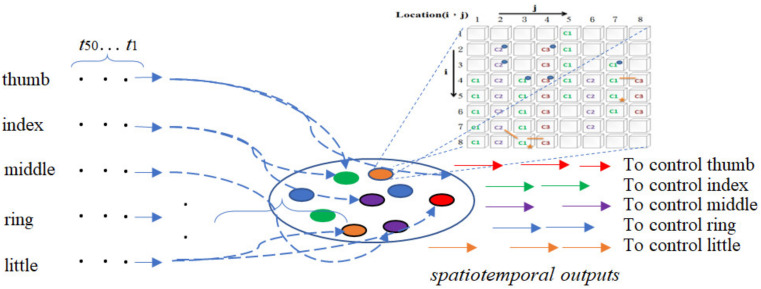
Input/output interface of one of the eight subnets.

**Figure 9 biomimetics-08-00076-f009:**
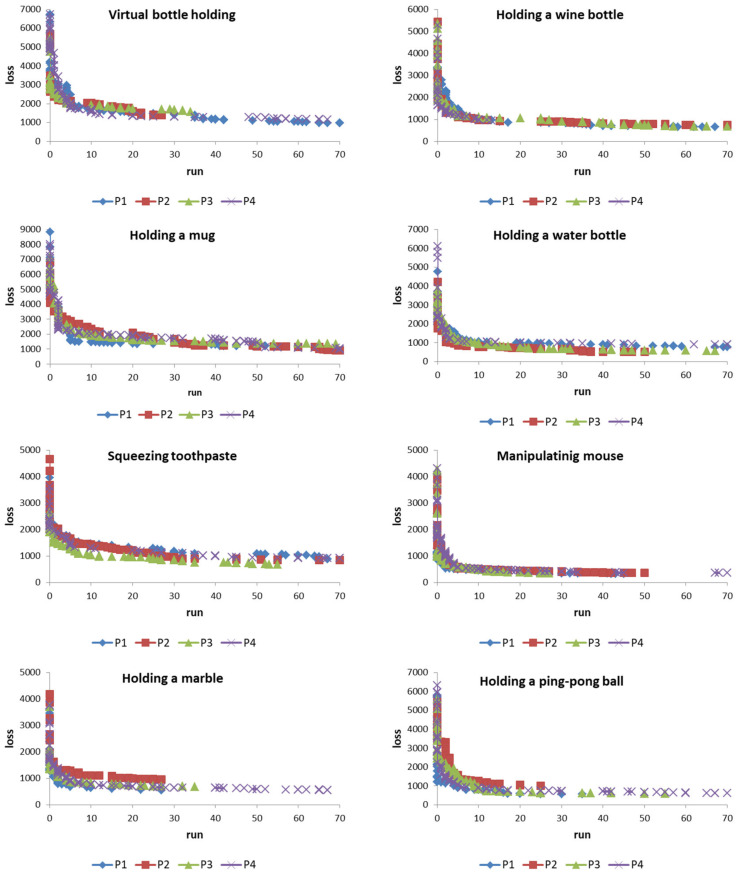
Learning progress of the ANM system on the conversion of 4 patient actions.

**Figure 10 biomimetics-08-00076-f010:**
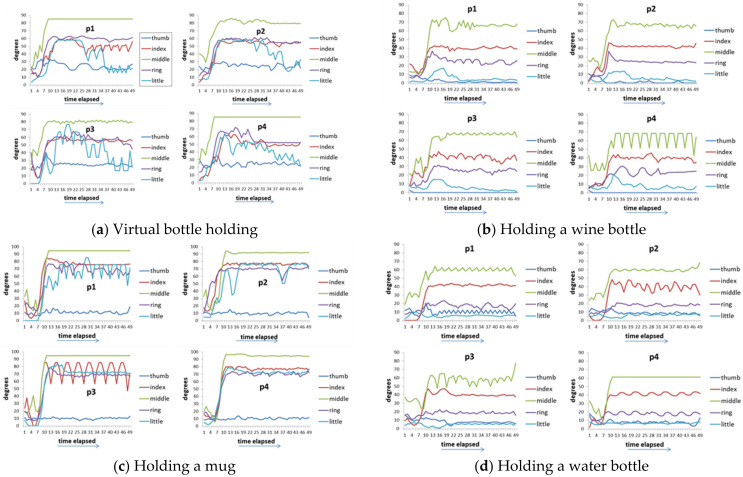
Time-series outputs of the five fingers of each patient with the control of the ANM system. (**a**) virtual bottle holding; (**b**) holding a wine bottle; (**c**) holding a mug; (**d**) holding a water bottle; (**e**) squeezing toothpaste; (**f**) manipulating mouse; (**g**) holding a marble; (**h**) holding a ping-pong ball.

**Figure 11 biomimetics-08-00076-f011:**
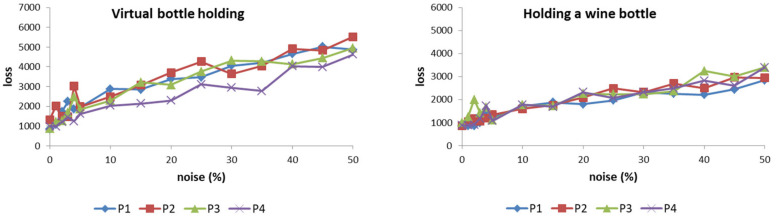
The loss value of the ANM system increases correspondingly as the degree of patient action changes increases.

**Table 1 biomimetics-08-00076-t001:** The $\bar{\mathrm{loss}}$ value before and after learning.

	P1		P2		P3		P4	
	Before	After	Before	After	Before	After	Before	After
Virtual bottle holding	27.0	3.9	22.8	5.3	22.0	3.5	27.0	4.0
Holding a wine bottle	18.2	3.4	21.8	3.4	21.4	3.9	21.1	3.5
Holding a water bottle	19.0	3.0	16.8	2.1	15.3	2.8	24.4	3.4
Holding a mug	35.3	4.7	27.2	3.5	28.7	4.8	32.1	2.7
Squeezing toothpaste	15.8	3.6	18.6	3.3	10.6	3.4	14.5	3.2
Manipulating mouse	23.2	2.3	21.7	3.0	22.4	2.4	25.3	1.9
Holding a marble	14.3	2.2	16.7	2.1	14.8	2.8	15.1	2.1
Holding a ping-pong	10.4	1.4	15.5	1.4	17.0	1.6	17.2	1.4

**Table 2 biomimetics-08-00076-t002:** The difference between patients (including between each patient and healthy people) before using the ANN system (in terms of $\bar{\mathrm{loss}}$ value).

Virtual Bottle Holding						Holding a Wine Bottle					
	p1	p2	p3	p4	Healthy		p1	p2	p3	p4	Healthy
p1	0	9171	5014	6712	6742	p1	0	6666	5746	6753	4550
p2		0	6600	7870	7686	p2		0	5282	5988	5441
p3			0	5840	6327	p3			0	5675	5359
p4				0	6370	p4				0	5270
healthy					0	healthy					0
Holding a mug						Holding a water bottle					
	p1	p2	p3	p4	healthy		p1	p2	p3	p4	healthy
p1	0	7327	8448	5811	8821	p1	0	4316	4379	3922	4756
p2		0	6533	4810	6794	p2		0	3854	4849	4310
p3			0	5367	7266	p3			0	4511	3835
p4				0	7533	p4				0	6112
healthy					0	healthy					0
Squeezing toothpaste						Manipulating mouse					
	p1	p2	p3	p4	healthy		p1	p2	p3	p4	healthy
p1	0	5765	6669	5759	3948	p1	0	5138	4763	5131	2601
p2		0	5551	6181	4661	p2		0	3768	4337	3879
p3			0	5129	2661	p3			0	4228	4259
p4				0	3625	p4				0	4307
healthy					0	healthy					0
Holding a marble						Holding a ping-pong ball					
	p1	p2	p3	p4	healthy		p1	p2	p3	p4	healthy
p1	0	7243	5313	6941	3587	p1	0	6324	5365	6372	5809
p2		0	5150	7094	4174	p2		0	4846	6227	5425
p3			0	7022	3698	p3			0	6562	5600
p4				0	3775	p4				0	6326
healthy					0	healthy					0

**Table 3 biomimetics-08-00076-t003:** The difference between patients (including between each patient and healthy people) after using the ANN system (in terms of $\bar{\mathrm{loss}}$ value).

Virtual Bottle Holding						Holding a Wine Bottle					
	p1	p2	p3	p4	Healthy		p1	p2	p3	p4	Healthy
p1	0	5044	3328	2992	862	p1	0	5802	5065	5923	549
p2		0	3958	3195	1325	p2		0	5225	4906	1135
p3			0	3152	884	p3			0	4568	1180
p4				0	989	p4				0	884
healthy					0	healthy					0
Holding a mug						Holding a water bottle					
	p1	p2	p3	p4	healthy		p1	p2	p3	p4	healthy
p1	0	4936	5364	5765	1178	p1	0	3490	4033	5454	719
p2		0	4630	5626	869	p2		0	3403	4513	643
p3			0	4598	1196	p3			0	4310	934
p4				0	684	p4				0	1140
healthy					0	healthy					0
Squeezing toothpaste						Manipulating mouse					
	p1	p2	p3	p4	healthy		p1	p2	p3	p4	healthy
p1	0	4677	6126	5405	905	p1	0	6999	6194	6834	338
p2		0	6429	5462	819	p2		0	6163	6936	358
p3			0	4916	854	p3			0	5297	389
p4				0	804	p4				0	356
healthy					0	healthy					0
Holding a marble						Holding a ping-pong ball					
	p1	p2	p3	p4	healthy		p1	p2	p3	p4	healthy
p1	0	6712	5149	7606	563	p1	0	5801	4826	5124	578
p2		0	5892	6409	519	p2		0	5097	6047	755
p3			0	7738	697	p3			0	6817	602
p4				0	528	p4				0	470
healthy					0	healthy					0

**Table 4 biomimetics-08-00076-t004:** Comparison of the learning performance of the ANN system and MLP (in terms of $\bar{\mathrm{loss}}$ value). The red color in the table highlights that MLP has a better performance than the ANN system.

	Virtual Bottle Holding		Holding a Wine Bottle		Holding a Mug		Holding a Water Bottle		Squeezing Toothpaste		Manipulating Mouse		Holding a Marble		Holding a Ping-Pong Ball	
	ANM	MLP	ANM	MLP	ANM	MLP	ANM	MLP	ANM	MLP	ANM	MLP	ANM	MLP	ANM	MLP
p1	861	1278	548	823	1177	947	760	919	905	639	338	1361	562	1510	578	519
p2	1325	1504	854	674	869	2061	643	934	818	401	358	984	518	1051	754	455
p3	884	1324	986	639	1195	1276	706	1053	853	757	388	1088	696	1121	601	493
p4	988	1295	883	680	684	712	841	1137	803	512	355	601	527	1623	469	592

## Data Availability

The data can be accessed found through the following link: https://drive.google.com/drive/folders/1xE5iKulffX4UruIUHSFEXn2zHWrrHYEi?usp=sharing.

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
