# Peer review of "Bridging the Finger-Action Gap between Hand Patients and Healthy People in Daily Life with a Biomimetic System"

_biomimetics, 2023, doi:10.3390/biomimetics8010076_

Round 1
Reviewer 1 Report
Authors presented a manuscript on “Bridging the Finger-Action Gap between Hand Patients and Healthy People in Daily Life with a Biomimetic System”. The manuscript is relevant and could be considered for publication after the following comments have been addressed.
1. Could the authors provide a confusion matrix that would help to easily visualise how well the ANM is able to effectively transduce each patient's movements into those of healthy people.
2. Could the authors compare their results with the state of the art in a tabular form. This will help to better see the novelty.
3. What is the accuracy of the glove used as data source and the integrity of the data used?
4. Some of the figures need to be improved. Example, the texts on Figure 7 and 10 is too tiny and the figure should be improved or presented differently.
Author Response
Authors presented a manuscript on “Bridging the Finger-Action Gap between Hand Patients and Healthy People in Daily Life with a Biomimetic System”. The manuscript is relevant and could be considered for publication after the following comments have been addressed.
- Could the authors provide a confusion matrix that would help to easily visualize how well the ANM is able to effectively transduce each patient's movements into those of healthy people.
[reply] Two tables were added into subsection 4.1 (Table 2 and Table 3). A paragraph was added into subsection 4.1 (lines 435-444). The paragraph added is as follows:
Table 2 shows the gap between patients (including between each patient and healthy people) before learning using the ANN system. From table 2 we can see that the gap between them (in terms of value) is quite large. However, after the learning of the ANN system, the gap between patients and healthy people has been significantly improved (Table 3). While the gap between patients did not improve as much as the gap between patients and healthy people, it did show some improvement. This result is due to the fact that ANM systems are designed to improve the gap between each patient and healthy people (not between patients). However, each patient still has its own sequelae that are different from others. In this way, they will all improve towards the same learning goal, but there is no doubt that each still has its own problems.
- Could the authors compare their results with the state of the art in a tabular form? This will help to better see the novelty.
[reply] A paragraph was added into section Discussion (lines 524-533). The paragraph added is as follows:
The ultimate goal of this research is not only to close the gap between the movement of patients and healthy people, but also to hope that this gap can change smoothly as the movement of patients varies. Under this premise, when discussing the pros and cons of a system, the combination of different time series should be considered instead of judging only by the results of a single time series. In response to this problem, the current system performance is analyzed under the assumption that the data is disturbed in different degrees of time and space. Therefore, this study designs two different experiments, one of which is about adaptive learning and the other is about fault tolerance. According to the author's understanding, the current research in this direction is quite limited.
- What is the accuracy of the glove used as data source and the integrity of the data used?
[reply] A paragraph was added into subsection 3.1 (lines 275-291). The paragraph added is as follows:
As mentioned earlier, the data used in this study were derived from the previous research of this team. In that study, we made an induction glove and then asked 30 people with healthy hand actions as well as 4 hand patients to do the above 8 actions (note: these subjects are all right-handed). We add here that all data collection in that study was done with care. Before the experiment and after a period of the experiment, the research team conducted a timely reliability and validity analysis on each sensor to ensure the accuracy of the data. The method we adopt is to use a protractor from 0 to 90, and then use 15 degrees as a measurement angle interval to make 6 angle lines. For each angle we bend the camber sensor to align with the predetermined angle and calculate the difference. For each angle value, we repeated the test 5 times to ensure that the angle value collected by the curvature sensor is correct. However, it is emphasized here that the focus of this article is not to analyze the data of these patients, but to achieve the purpose of making actions similar to healthy people through autonomous learning. Therefore, this study here only hopes to present the data of the patients without further analysis of these data. Basically, if it can be proved that there are some differences in the data of these patients and the ANM system can be used to narrow this gap. This is one of the main purposes of using this data glove in this study.
- Some of the figures need to be improved. Example, the texts on Figure 7 and 10 is too tiny and the figure should be improved or presented differently.
[reply]
The problem that the text in Figure 7 and Figure 10 is too small has been corrected. In addition, other figures are also moderately corrected.

Reviewer 2 Report
This paper tries to investigate an important problem in rehabilitation robots, that is, how to transfer the designed algorithm to each individual patient. Some detailed comments are given as follows:
1. The introduction is not organized in a logic way and therefore, I can not see the direct connection from the previous work to the novelty in this work. Th authors mentioned that the proposed system have an autonomous learning capability. However, this is over-rated compared with contribution of this paper.
2. There are too many grammar errors and all the figures should be improved.
3. For the control network, is this a totally new network or is this structure inspired from someone else's work? This is very important and this part should be emphasized.
4. Many related work from the community of robotic grasping and robotic hand could be potentially useful.
Author Response
This paper tries to investigate an important problem in rehabilitation robots, that is, how to transfer the designed algorithm to each individual patient. Some detailed comments are given as follows:
- The introduction is not organized in logic way and therefore I cannot see the direct connection from the previous work to the novelty in this work. The authors mentioned that the proposed system have an autonomous learning capability. However, this is over-rated compared with contribution of this paper.
[reply] A paragraph was added to the section Introduction (lines 95-111). The paragraph added is as follows:
This study hopes to establish a system with an autonomous learning capability that can provide some kind of assistance to users with special needs in a self-adjusting way. The ANM system is a biologically motivated system that captures the close structure/function relationship of biological systems [26-27]. This research team has previously successfully demonstrated that it has sufficient autonomous learning ability to learn to control the walking of a quadruped robot [28], the serpentine motion of a snake-like robot [29], and the human-like rehabilitation movement of a robot arm [30]. These actions are mainly achieved by adjusting and controlling the rotation angle and starting time of different motors by the ANM system. Different from the above-mentioned research, this study attempts to superimpose the output signals of the ANM system to generate a wave pattern that controls finger bending. Our goal is to bridge the gap in daily life finger movements between hand patients and healthy individuals through the self-learning capabilities of the ANN system that we demonstrated previously (i.e., building a customizable prosthetic finger control system). However, it should be added here that the current research stage of this study is that the control of the prosthetic finger system is verified under a simulated platform, rather than under a physical operating environment.
- There are too many grammar errors and all the figures should be improved.
[reply] Grammatical errors and figure-related issues have been corrected.
- For the control network, is this a totally new network or is this structure inspired by someone else's work? This is very important and this part should be emphasized.
[reply] A paragraph was added to subsection 2.4 (lines 205-209). The paragraph added is as follows:
Although current technology is quite developed and advanced, however, how we form and store memories is still a mystery that has not been fully solved. It is generally believed that the hippopotamus is the main location of long-term memory. The control structure of this study is motivated by the parallel, sequential, and hierarchical structure of hippopotamus memory structure. Some related literature can also be found in [35].
- Many related works from the community of robotic grasping and robotic hand could be potentially useful.
[reply] A paragraph was added to the section Introduction (lines 65-74). The paragraph added is as follows:
In the field of finger-controlled robotics research, there are currently many types of robotic arms with grasping (grasping) capabilities [18]. If divided from the source of power, it includes vacuum, pneumatic, electric and hydraulic. In addition to the different power sources mentioned above, the arms may also have different numbers of jaws. At present, most robots are designed with hard materials, but there are still some robotic arms that choose to use relatively soft materials [19]. Moreover, generally speaking, the control of the movement of the robotic fingers is accomplished through close contact, but there are also a group of scholars who control the movement of the fingers by remote control [20]. Each design has special features suitable for the various field of application and for different needs.

Round 2
Reviewer 1 Report
In general authors seem to have carried out cosmetic revision of the work and have not adequately responded to the previous comments from reviewer. For instance, the texts in Figure 10 is still very small and difficult to read. Also, the sub-sections of Figure 10 should be labelled as a, b, c d, etc. Furthermore, authors were required to compare their work with the state of the art in a tabular form with key parameter. This was not provided.
Author Response
[Comment] In general authors seem to have carried out cosmetic revision of the work and have not adequately responded to the previous comments from reviewer. For instance, the texts in Figure 10 is still very small and difficult to read.
[Reply] Thanks again for the valuable comments. The author reworks Figures 5, 7, 9, 10, 11.
[Comment ] Also, the sub-sections of Figure 10 should be labelled as a, b, c d, etc.
[Reply] As suggested, Figure 10 is labelled as a, b, c d, etc.
[Comment] Furthermore, authors were required to compare their work with the state of the art in a tabular form with key parameter. This was not provided.
[Reply] The work of this study is compared with MLP. The author adds one paragraph (lines 467-486) to explain the results of comparison. Also, a new table (Table 4) is added to show the work of the ANM system and MLP. Please see the pdf file.

Reviewer 2 Report
No further comments for this submission and the quality of the figures in the final submission should be improved.
Author Response
The author thanks the time and valueable comments of both reviewers. The grammatic and spelling errors have been corrected.
